# Measurement report: Aerosol and cloud nuclei properties along the Central and Northern Great Barrier Reef – Impact of continental emissions

E. Johanna Horchler[1,*], Joel Alroe[1], Luke Harrison[2], Luke Cravigan[1], Daniel P. Harrison[2,3], Zoran D. Ristovski[1]

[1]International Laboratory for Air Health and Quality, Queensland University of Technology, Brisbane, 4001, Australia
[2]National Marine Science Centre, Southern Cross University, Coffs Harbour, 2450, Australia
[3]School of Geosciences, University of Sydney, Sydney 2006, Australia
[*]Now at: Department of Chemistry, Faculty of Natural Science, Aarhus University, Aarhus C, 8000, Denmark

*Correspondence to*: Eva Johanna Horchler (eva.horchler@qut.edu.au)

**Abstract.** The frequency of coral bleaching events increased during the past decade in the Great Barrier Reef due to climate change, and rising ocean temperatures. Recent work has demonstrated that enhancing local-scale cloud albedo can reduce the sea surface temperatures in this region. However, little research has been done on variations in the aerosol properties, as well as aerosol-radiation and aerosol-cloud interactions over different regions of the Great Barrier Reef, which is critical for predicting the potential for Marine Cloud Brightening climate forcing on a local or regional scale. Here, we examined trends in the aerosol population in terms of their physical and cloud condensation nuclei properties during a research cruise in the Central and Northern Great Barrier Reef. Overall higher particle number concentrations, cloud condensation nuclei number concentrations, and cloud condensation nuclei activation ratios were observed during periods where the air masses passed over the continent prior to reaching the research vessel, despite lower hygroscopicity parameters. We suggest that organics contribute considerably to cloud condensation nuclei number concentrations in this region of the Great Barrier Reef, which highlight the important role of additional emissions from inland Queensland. As well as the total aerosol number concentration, precipitation history along the back-trajectory impacted cloud condensation nuclei number concentrations. These results represent a first step towards building a climatological understanding of aerosol and cloud condensation nuclei properties over the Great Barrier Reef during summertime, a region and season where no observations have been previously reported.

## 1 Introduction

The Great Barrier Reef (GBR) covers an area of 344,000 km² and stretches over 2300 kilometres in latitude along Eastern Australia's coastline. It is the world's biggest living structure, and home to over 600 coral varieties (Mallet et al., 2016; Hock et al., 2019; Condie et al., 2021).

Thermal stress disrupts the symbiotic relationship between corals and some algae species, causing coral bleaching (Berkelmans et al., 2004; Hughes et al., 2017). Higher levels of coral mortality are typically the result of prolonged and severe coral

bleaching (Hughes et al., 2017). In addition, the frequency of bleaching events has increased because of climate change, and rising ocean temperatures (Hughes et al., 2017; Hock et al., 2019; Stuart-Smith et al., 2018; Condie et al., 2021), with 4 out of the 6 mass bleaching events over the GBR occurring in the past 7 years (1998, 2002, 2016, 2017, 2020, and 2022) (Hughes et., 2017; Condie et al., 2021, GBRMP, 2023). Under future climate change scenarios, this phenomenon is likely to intensify and become more frequent (Berkelmans et al., 2004). Even though reduction in greenhouse gas emissions have been attempted and climate interventions have been implemented in recent years to mitigate the effects of climate change on the GBR, emission reductions are no longer sufficient to ensure the survival of the GBR as we know it. A loss of 4.4 % of total coral cover is observed on average every decade, with an increase in the loss rate predicted in the next few decades (Condie et al., 2021).

Local-scale cloud cover has been demonstrated to affect sea surface temperatures (SSTs) on the GBR and appears to be important in mitigating or exacerbating coral bleaching through its effect on the oceans heat budget (Leahy et al., 2013; Zhao et al., 2021; Zhao et al., 2022). Knowledge on the aerosol number concentration, chemical composition, and size, as well as aerosol-cloud interactions over the GBR is required to estimate aerosol effects on radiation reaching and leaving the sea surface. Aerosol-cloud interactions refer to the aerosol's ability to act as cloud condensation nuclei (CCN) influencing cloud microphysical properties (Murphy et al., 1998; Andreae and Rosenfeld, 2008; Zhao et al., 2023). Modelling these effects is difficult due to the variations in numerous input parameters, as well as the complexity of interdependent processes involved (Andreae and Rosenfeld, 2008; Cropp et al., 2018; Cravigan et al., 2020; Fiddes et al., 2021), and measurements of such over the GBR. As a result, aerosol-cloud interactions constitute by far the biggest contributor to overall uncertainty in radiative forcing modelling (Masson-Delmotte et al., 2021).

Aerosols over the marine environment can be locally derived or transported from elsewhere. Marine sources of aerosols and therefore CCN are mainly from sea spray or from secondary marine sources. Sea spray aerosol (SSA) is formed through bubble bursting, and are primarily composed of sodium chloride, other inorganic salts and some organic compounds produced by marine biota (Heintzenberg et al., 2000; Cravigan et al., 2020). Secondary marine aerosols (SMA), a mixture of non-sea salt (nss) sulfates and organic compounds are generated from the oxidation of aerosol precursors (Quinn et al., 2017; Jackson et al., 2018). In addition, land outflows carrying terrestrial emissions might change aerosol composition and loading over the GBR. Additionally, precipitation and particle entrainment from the free troposphere (FT) have been previously found to alter the particle size distributions and CCN number concentration in marine environments (Tunved et al., 2004; Andreae and Rosenfeld, 2008; Quinn & Bates, 2011; Clarke et al., 2013; Atwood et al., 2017; Khadir et al., 2023).

The Reef Restoration and Adaptation Program (RRAP) is a collaboration of Australia's leading experts to create a suite of innovative and targeted measures to help preserve and restore the GBR (RRAP2023). Marine cloud brightening (MCB) is being evaluated as a potential technique to reduce the heat stress on corals (Harrison et al., 2019; Harrison, 2024). In order to assess the applicability and likely effectiveness of MCB in the GBR region it is important to understand the sources and sinks of CCN within this region. To attempt to address this question and as a part of the RRAP, the aerosol properties and the aerosol contribution to CCN were assessed in a transect voyage conducted in February and March 2022, covering parts of the central (~16.4° S to ~20° S) and the northern (~10.4° S to 16.4° S) sector of the GBR.

## 2 Materials and Methods

### 2.1 Survey area

The data was collected over parts of the Northern and the Central GBR from 26 February to 13 March 2022 in the Cloud-Cube container aboard the Riverside (RV) Magnetic, covering an area from 145° E to 147° E in longitude and 14° S to 19° S in latitude. The voyage track is shown below in Fig. 1. Despite its marine environment, the GBR can receive atmospheric outflows from land, carrying terrestrial and anthropogenic emissions possibly leading to greater aerosol loading and different aerosol composition (Chen et al., 2019; Fiddes et al., 2022; Hernandez-Jaramillo et al., 2024). The northern half of the Central GBR and the Northern GBR are characterised by a continuous sequence of interconnecting shelf-edge reefs, as well as massive, plentiful reefs, which are located closer to the shore than in other regions of the GBR and likely receive these outflows with increased strength and frequency (Fig.1). Townsville and Cairns are both cities with over 500,000 inhabitants, and, in addition to industry, feature a port, a marina, and an airport. Depending on the wind direction, emissions from these can thus be transported to the GBR. Air masses passing over the large areas of rainforest, particularly in Queensland's northern region, might carry an increased contribution of biogenic organic aerosols. Furthermore, regional, and long-range transport of biomass burning emissions in Northern and Eastern Australia might additionally influence the aerosol load and properties over the GBR (Mallet et al., 2017; Milic et a., 2017). In view of these potential influences to atmospheric conditions over the GBR, air mass back-trajectories were generated using the Hybrid Single-Particle Lagrangian Integrated Trajectory (HYSPLIT) dispersion model (Stein et al., 2015), for all points along the ship track, to assist in attributing changes in aerosol properties to potential sources.

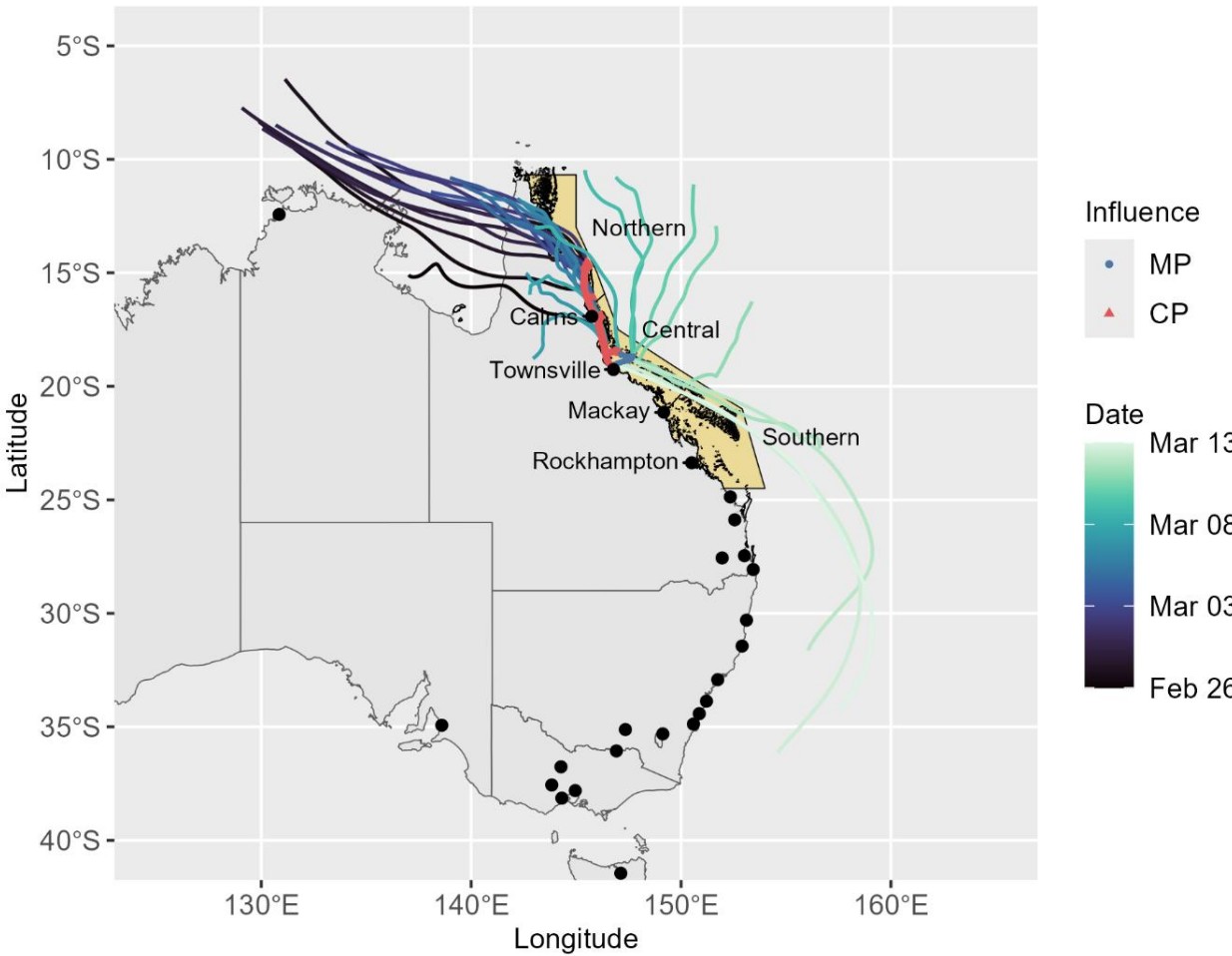

**Figure 1:** Voyage track for the survey in February and March 2022 aboard the Riverside Magnetic. The ship path is colour-coded according to whether the data shows a "continental" or "marine" influence based on the three-day back-trajectory pathways, with continental periods (CP) in red triangles, and marine periods (MP) in blue points. The yellow area shows the Great Barrier Marine Park, separated into the "Northern", "Central", and "Southern" section, and the black area marks the GBR, both provided in the gisaimsr package by the Geoscience Australia (GA) and the Great Barrier Reef Marine Park Authority (GBRMPA). Additionally, the three-day air mass back-trajectories at 12-h intervals (12 am and 12 pm local time), with a trajectory starting height of 10 m above mean sea level (sampling inlet height) are presented, colour-coded by the date. The ship location marks the start point of each trajectory. The continental boundaries were obtained from the ozmaps package (doi:10.32614/CRAN.package.ozmaps). Major cities were provided in the maps package (http://www.populationmondiale.com), with those with a population larger than 50,000 marked as black points and the ones along the GBR labelled.

## 2.2 Cloud-Cube mobile laboratory

The Cloud-Cube (Fig. S1) is a self-contained multi-instrument laboratory intended as a portable measurement station, housed in a small 8' shipping container with dimensions of 1700 x 1980 x 1835 mm and weighing less than 1000 kg. It was designed for ease of installation on the deck of any ship, and requires limited equipment maintenance, such that it can be operated by

minimal scientific staff or even the vessel crew. The container was divided into two sections: one for the instruments, which was constructed as an insulated air-conditioned chamber to keep the equipment at a moderate temperature, and another for pumps and other equipment, which had additional ventilation. A quarter of the container is sealed off from the remainder with drainage and opens to the sky by way of a hatch in the ceiling allowing for vertically pointing instruments such as the LiDAR ceilometer (CL-61, Vaisala) to operate from inside the container. The container was situated in the front part of the ship. Air samples are captured through a total suspended particulate sampling inlet (URG-2000-30DG, URG) mounted on a 4 m long 3/8-inch copper tube extending vertically from a corner of the Cloud-Cube container, with the inlet approximately 9 m above the ocean's surface.

The Particle Loss Calculator (von der Weiden et al., 2009) was used to estimate the size dependent particle inlet efficiency (Fig. S2), which was then utilised to correct particle size distribution observations.

The ambient relative humidity (RH) was measured by a Gill weather station (Gill MaxiMet GMX501, Gill Instruments Ltd.). A Nafion dryer (MD700, Perma-Pure LLC) was used to dry the sample flow, and the sample line RH was monitored continuously by the Scanning Electrical Mobility Spectrometer (SEMS 2100, Brechtel Manufacturing Inc.) inbuilt sample line RH sensor.

## 2.3 Instrumentation

Total number concentrations of particles with diameters $\geq 7$ nm was monitored using a stand-alone mixing Condensation Particle Counter (mCPC 1720, Brechtel Manufacturing Inc.).

Number size distributions of particles with diameters between 10 nm and 10 μm were measured with a Scanning Electrical Mobility Spectrometer (SEMS 2100, Brechtel Manufacturing Inc.) combined with a mCPC 1720, and an Aerodynamic Particle Sizer (APS 3321, TSI Inc.).

A Cloud Condensation Nuclei counter (CCN-100, Droplet Measurement Technologies) was provided to continuously monitor the number of CCN and was operated in scanning mode (0.1 %, 0.2 %, 0.3 %, 0.5 %, 0.7 % supersaturation (SS) for 10 minutes each). Only CCN with a SS of 0.3 % will be adopted in this study, if not noted otherwise, as an averaged in-cloud supersaturation of approximately 0.29 % for the continental period (CP) and 0.36 % for the marine period (MP) could be observed (classification into CP and MP is provided in Section 2.5). The Supplementary Information (SI) includes the procedure for estimating in-cloud supersaturation. Given that the particles most likely also included a mixture of other non-sea salt sulfates, organics, and a small number fraction from sea spray aerosol a supersaturation of 0.3 % appears to be suitable to assess CCN properties at cloud-relevant SS levels.

Meteorological data were collected by the Gill weather station (Gill MaxiMet GMX501, Gill Instruments Ltd.).

A detailed description of the instruments' configuration and the data analysis is given in the SI. Figure S3 depicts the sample flow arrangement for the equipment within the Cloud-Cube.

## 2.4 Pollution Identification

The GBR is a major shipping route and is frequently influenced by anthropogenic sources from the nearby Australian coastline. As previously mentioned, continental outflow might increase the aerosol load and affect the composition of the aerosol. Periods with continental influence should not be omitted since the impact of these additional emissions on the aerosol properties are addressed in this study. To account for these regional influences while removing occasional contamination by ship exhaust from the RV Magnetic, only data within a 180° frontal arc (270° to 90°) relative to the ship's heading, and wind speeds larger than 5 m s$^{-1}$ were included. Short periods of rapid fluctuations and considerable increases in aerosol number concentrations were attributed to self-sampling of the ship's engine exhaust and additionally removed from the data set.

## 2.5 Air mass categorisation and meteorological conditions

Three day back-trajectories were estimated using the HYSPLIT model to identify periods that may have been influenced by continental emissions. The trajectory starting height was set to 10 m above sea level, and the meteorological data were obtained from the Global Data Assimilation System (GDAS) at a 1° spatial resolution. A more detailed description of the back-trajectory calculation is provided in the Section S1 in the SI. Figure 1 shows the averaged three-day ensemble trajectories at 12-h intervals (12 am and 12 pm local time). Any air masses that passed over land within these three days prior to reaching the ship showed a clear land signature and were thus classified as "continental" periods (CP), all others as "marine" periods (MP). Continental boundaries were derived from the ArcGIS Hub (https://hub.arcgis.com/datasets/esri::world-continents/explore?location=-0.147065%2C0.000000%2C1.82&showTable=true). However, it should be noted that the established criteria for the CP or the MP might not adequately reflect the given conditions, such as long-range transport, local wind systems, or limitations in the accuracy of the modelled back-trajectories. Thus, the distinction between marine and continental air masses is more of a qualitative guide than a guarantee of their origin.

Furthermore, locations of fire hotspots from three days before the sampling period in the survey area were obtained from the Sentinel Hotspots system (https://hotspots.dea.ga.gov.au/acres/sentinel) using the MODIS (Moderate Resolution Imaging Spectroradiometer) sensors and VIIRS (Visible Infrared Imaging Radiometer Suite) sensors. Only hotspots with a confidence level higher than 50 % were considered in the analysis.

The temperature ranged from 23.28° C to 32.38° C over the course of the measurement period, with an average temperature of 28.09±1.12° C (median 28.28° C). The RH ranged from 43.64 % to 90.64 %, with an average value of 75.54±5.57 % (median 76.64 %). A time series of the temperature and RH for the entire period is provided in Fig. S4.

All data presented in the following have been cleaned for ship exhaust according to the method presented in Section 2.4

# 3 Results and Discussion

## 3.1 Marine versus continental air masses

The ability of aerosols to act as CCN is affected by the aerosol concentration, their size distribution and the chemical composition (Andreae and Rosenfeld, 2008).

CCN number concentrations were highly variable throughout the measurement period with larger fluctuations noticed during CP, as shown in Fig. 2. The average CCN number concentration for the MP was $194\pm72$ cm$^{-3}$ (median 196 cm$^{-3}$), with individual values ranging from 31 cm$^{-3}$ to 374 cm$^{-3}$. For the CP, the average was $370\pm93$ cm$^{-3}$ (median 358 cm$^{-3}$), with individual values ranging from 112 cm$^{-3}$ to 789 cm$^{-3}$, which are within the range of previous observations at costal marine sites (Schmale et al., 2018). The higher concentrations in the CP reflect that continental outflows were a contributor to CCN number concentrations in this region of the GBR.

The tendency for aerosols produced in oceanic environments to be hygroscopic enhances their ability to activate as CCN (Boucher et al., 2013). Conversely, the hygroscopicity parameter $\kappa$ can be used to provide information about the aerosol composition (Petters and Kreidenweis, 2007). For example, ammonium sulfate, which has a hygroscopicity parameter $\kappa$ of roughly 0.53 (Petters and Kreidenweis, 2007), is used as a surrogate for nss-sulfates. Ambient nss-sulfates are commonly internally mixed with organic species, which are hence lowering the $\kappa$ value (Cravigan et al., 2020). Zieger et al. (2017) points out a hygroscopicity parameter $\kappa$ of 1.1 for inorganic sea salt. Secondary organic aerosols have a $\kappa \cong 0.1 - 0.2$, while biomass-burning aerosols with primarily soot particles show a $\kappa$ of 0.01, and from grass burning of 0.55 (VanReken et al., 2005; Prenni et al., 2007). In organic dominated marine tropical environments with continental influence, Irwin et al. (2011) reported $\kappa$ values of 0.05 to 0.37 for a SS range between 0.73 % and 0.11 %, respectively. Atwood et al. (2017) found a $\kappa$ of 0.22 for a mixture of marine and biomass burning aerosols in the South China Sea/East Sea. The $\kappa$ values measured in our study as shown in Fig. S5 for MP and CP at various SS levels are thus consistent with findings for marine environments with organic contribution. For all SS levels, nss-sulfates and organics tended to dominate the activated aerosol fraction, with the organic percentage being slightly larger in the CP than in the MP. This is expected given that continental outflows of both anthropogenic and biogenic origin contain a considerable amount of organic material.  However, significant variations especially in continental $\kappa$ were discovered, with an apparent greater number of outliers, hinting an additional contribution of SSA to the CCN number concentrations during the CP. This in turn leads to an increase in the mean $\kappa$ compared to the median $\kappa$ in the CP (Fig. 2). With increasing SS, more hydrophobic aerosols such as organics activate, lowering the $\kappa$. In general, the low $\kappa$ values suggest that organics contribute considerably to CCN in this region of the GBR during both the CP and the MP. No indicative difference in median $\kappa$ was observed between the MP and the CP at cloud-relevant SS (Fig. 2). The $\kappa$ values for the MP varied from 0.037 to 0.629, with an average of $0.188\pm0.081$ (median 0.192). For the CP, the $\kappa$ values ranged from 0.006 to 1.383, with an average of $0.293\pm0.267$ (median 0.172). Sea spray enriched in organics was previously found to have low $\kappa$ values and greater CCN number concentrations (Andreae and Rosenfeld, 2008; Ovadnevaite et al., 2011; Ovadnevaite

et al., 2017). This could explain our findings that the CCN number concentration in the CP is higher than in the MP, despite the somewhat lower κ.

Similar to CCN number concentrations, CN number concentrations are higher in the CP and vary more than in the MP (Fig. 2). The average CN number concentration was 631±209 cm$^{-3}$ (median 559 cm$^{-3}$), with values ranging from 308 cm$^{-3}$ to 2468 cm$^{-3}$ for the CP, and 387±101 cm$^{-3}$ (median 386 cm$^{-3}$), ranging from 139 cm$^{-3}$ to 728 cm$^{-3}$ for the MP. The number concentrations in this study are thus within the range of previously reported number concentrations of approximately 400 to 800 cm$^{-3}$ for partially continental influenced marine environments covering the same latitudinal band (Heintzenberg et al.,

2000), as well as modelled total aerosol number concentrations for marine background of approximately 150 to 400 cm$^{-3}$ (Mann et al. 2010). The fact that averaged marine concentrations are at the upper concentration range of clean marine environments further indicate a non-negligible continental contribution.

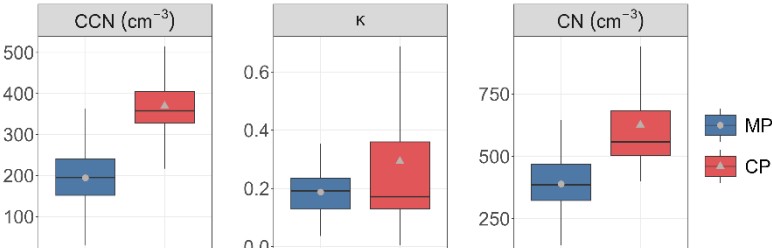

**Figure 2:** CCN number concentration, hygroscopicity parameter κ, and CN number concentration for the MP (blue) and the CP (red) at 0.3 % SS. The black horizontal line in a box displays the median of the individual data. The lower and upper hinges represent the 25th and 75th percentiles. The upper and lower whiskers extend from the hinge to the largest or smallest measured values, respectively, but not more than 1.5 times the difference between the 25th and 75th percentiles. The mean is shown as grey points for the MP and grey triangles for the CP.

The modest changes in median κ values between the MP of 0.192 and the CP of 0.172 suggest that discrepancies in CCN number concentrations could not be attributed to differences in the hygroscopicity. The ratio between median continental CN number concentrations and median marine CN number concentrations of 1.45 is lower than the ratio between median continental CCN number concentrations and median marine CCN number concentrations of 1.83. Therefore, the differences

in CCN number concentrations between the CP and the MP cannot be explained solely by the differences in CN number concentrations. As previously stated, and addressed in Andreae and Rosenfeld (2008), aerosol size distributions, which are impacted by the cloud microphysical conditions in the back-trajectory, are a key parameter determining the CCN number concentrations. Figure 3 presents average aerosol size distributions for MP and CP, including up to three log-normal fitted modes (dotted orange, dashed red, and dot-dashed blue) and the overall log-normal fit (solid black). The study focuses on

particle sizes smaller than 500 nm due to the generally low particle number concentrations for particles larger than that.

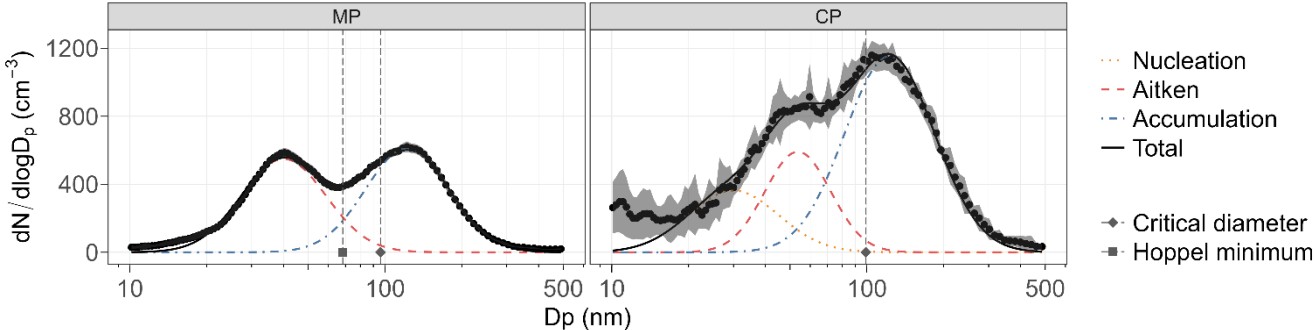

**Figure 3:** Average aerosol number size distributions for the MP, and the CP. The lines represent lognormal fits for an example representative nucleation mode (dotted orange), Aitken mode (dashed red), accumulation mode (dot-dashed blue), and total (solid black). The grey-shaded area shows the standard deviation. The dashed grey vertical line at the diamond-shaped point represents the critical diameter ($D_{crit}$), whereas the dashed grey vertical line at the square-shaped point represents the Hoppel minimum.

During the MP, a bimodal particle number size distribution, where the Aitken mode (dashed red) and accumulation mode (dot-dashed blue) were separated by the Hoppel minimum, could be observed. This is consistent with cloud-processed aerosols commonly seen in marine environments (Hoppel et al., 1986; Heintzenberg et al., 2000; Seinfeld and Pandis, 2006). During the CP, an additional mode at around 30 nm (nucleation mode, dotted orange) was observed. Since the CP represents an average of many different continental outflow events, the Hoppel minimum is not clearly defined in the total lognormal fit (solid black line in Fig. 3). When examining the individual data points, it appears that the CP has a Hoppel minimum at 66.1 nm, which coincides to the MP's minimum at 68.3 nm. The critical diameter ($D_{crit}$) and the Hoppel minimum for MP and CP are listed in Table S1 in the SI. Differences between the Hoppel minimum and $D_{crit}$ are most likely caused by the fact that the Hoppel minimum is driven by the SS during cloud processing along the back-trajectory, whereas $D_{crit}$ from the CCN is determined by the set SS during our measurements. The higher number concentrations in the CP compared to the MP mainly result from increased concentrations of particles in the accumulation mode and, to some extent, from increased concentrations in the nucleation mode. However, the latter show large standard errors, supporting previous assumption that the air masses in this part of the GBR primarily consist of a more homogeneous background air impacted by individual nucleation episodes. This can also be observed in the particle size distribution displayed in Fig. S6. The displayed averaged nucleation mode thus represents the mean of several observed nucleation events.

CCN number concentrations at cloud-relevant SS levels generally follow the particle number concentration in the accumulation mode. Due to their size, particles in this mode will activate regardless their compositions, whereas particles in the nucleation mode are expected to only play a minor role for CCN considering their small size, restricting them to activate as CCN. This could explain why the CCN number concentrations in the CP were found to be higher than those in the MP.

Cravigan et al. (2020) observed that increased organic concentrations in the ocean's surface layer may result in a shift towards larger particle diameters, thus resulting in higher CCN number concentrations. Alternatively, liquid-liquid phase separation of organic-rich aerosol particles could have contributed to the enhanced CCN number concentration (Ovadnevaite et al. 2017).

Both support the earlier claim that aerosol populations enriched in organic compounds can have greater CCN number concentrations despite lower κ values.

Because the displayed particle size distributions are an average and might not represent the particle size distributions of each area along the GBR, Fig. 4 shows hourly variations in the CN, and CCN number concentrations to help disentangle the impact of various sources on aerosol properties.

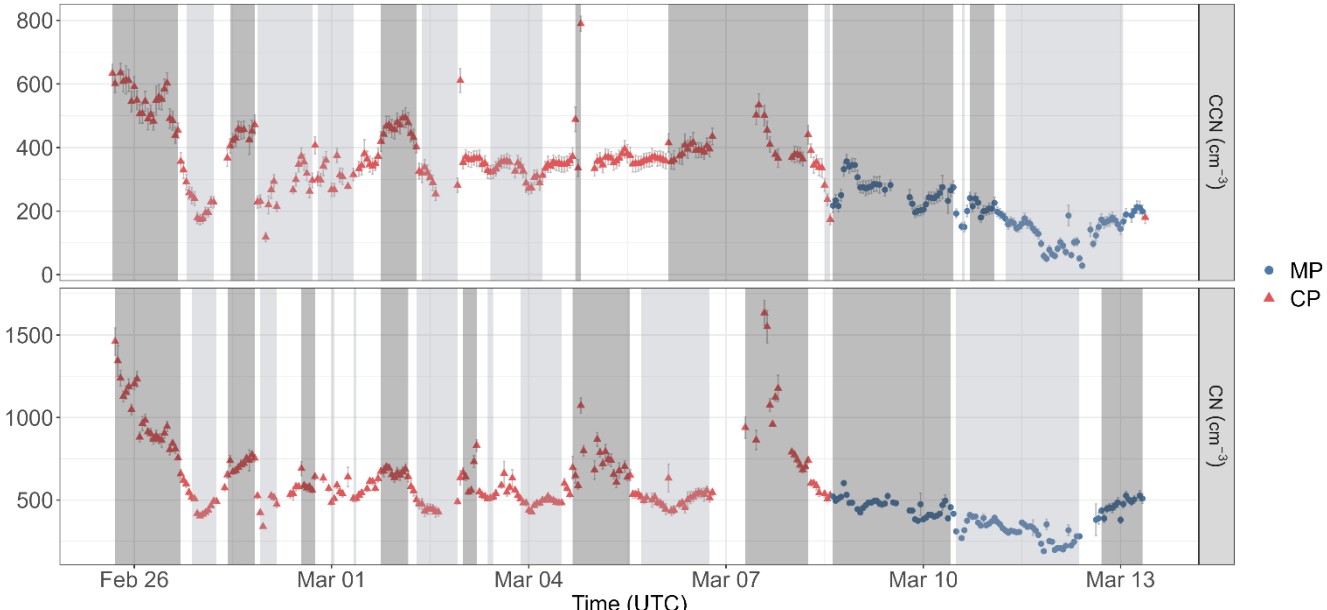

**Figure 4:** Hourly time series of CN, and CCN number concentrations at 0.3 % SS for the MP (blue points) and the CP (red triangles), including the standard deviation in grey. The dark grey-shaded areas represent periods with values more than 10% above the median, whilst the light grey-shaded areas represent periods with values more than 10% below the median.

The CN number concentrations reveal large spikes, particularly during the CP, which could be attributed to pollution exhaust from nearby ships, or additional continental outflows of both anthropogenic and biogenic origin. Despite these individual concentration spikes the CCN number concentration trend correlates positively with the CN number concentration. As distance from the shore increases after 07 March 2022, the CN and CCN number concentrations decrease. This suggests that individual nucleation bursts caused by increased emissions of volatile organic compounds from the dense rainforests north of Cairns or industrial emissions are better reflected in measurements collected nearshore. While bushfires are also a common source of increased number concentrations in continental outflows from northern Australia (Mallet et al., 2017; Milic et al., 2017), the Sentinel Hotspot System did not reveal any fires in the vicinity of the air mass back-trajectories. Bacteria, pollen, spores and plant debris might contribute to CCN as they are prevalent in the Aitken and accumulation modes. SOA formed through the oxidation of volatile organic compounds (VOCs) emitted by the rainforest might be an additional source of CCN, provided

they have grown to a size sufficient to be CCN active. Fresh fossil fuel from diesel engines peaks in the accumulation mode, whereas petrol engines exhibit smaller particle diameters predominantly in the Aitken mode. Aging causes the particles to grow in size and become more hygroscopic, adding to CCN.

In addition to continental outflows, prior research revealed that a considerable fraction of CCN in the marine boundary layer (MBL) result from particle nucleation in the FT and/or particle entrainment from the FT into the MBL (Tunved et al., 2004; Andreae and Rosenfeld, 2008; Quinn & Bates, 2011; Clarke et al., 2013; Khadir et al., 2023). These particles are predominantly internal mixtures of organic and inorganic compounds from diverse sources (Andreae and Rosenfeld, 2008), modulated by several non-precipitation cloud processing steps and are already within a size range that is CCN active. Entrainment into the

MBL occurs in the order of up to several days. Hence, for sampling locations such as the one in our study, air masses likely contain both locally formed and FT entrainment aerosol particles. Furthermore, precipitation potentially alters the CCN number concentration by lowering particle concentration in the accumulation mode (Tunved et al., 2004; Atwood et al., 2017; Khadir et al., 2023).

To determine the effects of the back-trajectory pathways, the fraction the back-trajectory spent in the FT, and precipitation on

CN number concentrations, aerosol size distributions, and composition, and hence eventually CCN properties, periods with a 10% deviation from the median CN or CCN number concentrations were identified (Fig. 4). The dark grey-shaded areas reflect periods with values more than 10% higher than the median, whereas the light grey-shaded areas represent periods with values more than 10% lower. The marked periods for the CN and CCN number concentrations are in good agreement, with only a few exceptions. The periods determined by the CN number concentrations will be chosen for further investigation. Each period

will be discussed individually, because of the different location and air mass origin implying different aerosol sources.

**3.2 High and low CN number concentration periods**

Figure 5 shows the aerosol number size distributions including $D_{crit}$ and, if present, the Hoppel minimum (top) and the three-day back-trajectories (bottom) for the seven continental high CN number concentration periods (HCP1-7). For improved visualisation, only data points every 12 hours along the back-trajectories are shown. Hourly resolved back-trajectories are

295 presented in Fig. S7. The aerosol size distributions and the three day back-trajectories for the continental low CN number concentration periods (LCP1-8) are shown in Fig. S8. The Hoppel minimum and $D_{crit}$ of all periods are additionally listed in Table S1.

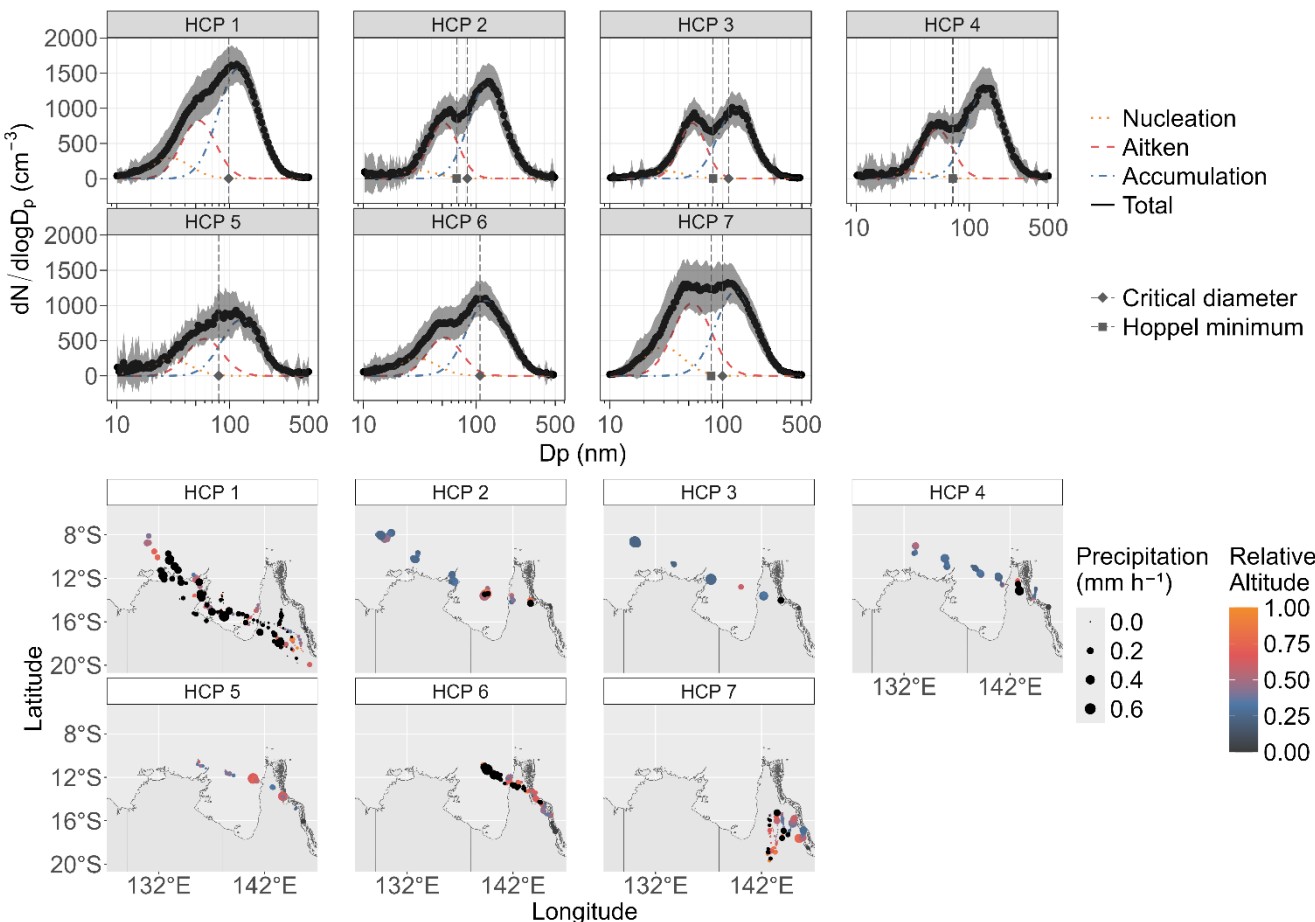

**Figure 5:** Average aerosol number size distributions (top) and the three-day back-trajectories (bottom) for the seven continental high CN number concentration periods (HCP1-7). The lines in the size distribution plots represent lognormal fits for the nucleation mode (dotted orange), Aitken mode (dashed red), accumulation mode (dot-dashed blue), and total (solid black). The dashed grey vertical line at the diamond-shaped point represents the critical diameter ($D_{crit}$), whereas the dashed grey vertical line at the square-shaped point represents the

305 Hoppel minimum. Back-trajectories are coloured by their altitude in respect to the planetary boundary layer height. Black data points represent air masses travelling through the free troposphere. The precipitation amount determines the size of the trajectory points. The ship location marks the start point of each trajectory. The GBR was provided in the gisaimsr package by the Geoscience Australia (GA) and the Great Barrier Reef Marine Park Authority (GBRMPA). The continental boundaries were obtained from the ozmaps package (doi:10.32614/CRAN.package.ozmaps).

The accumulation mode generally dominates in the particle size distribution. When the emission sources are located inland and mainly travelled over the Great Dividing Range in the Townsville and Cairns hinterlands (HCP1 and HCP7), the contribution to aerosols in the nucleation and Aitken modes is greater compared to when the air masses originate in the Arafura Sea. These are most likely local emissions from mines, farmland, industry, and ports near the major cities along the coast.

Elevated nucleation mode concentrations for air masses originating in the Arafura Sea and passing over the Cape York Peninsula suggest that these particles are most likely SOA formed by the oxidation of VOCs emitted by the rainforest, as there

are no large cities with industrial emissions in this region. However, without additional chemical composition analysis, it is impossible to clearly determine the emission source.

The Spearman correlation coefficients for a correlation significance level $p < 0.05$ between the CCN number concentration,
CN number concentration inferred from the total fit, number concentration in the accumulation mode, number concentration in the Aitken mode, number concentration in the nucleation mode, averaged precipitation along one trajectory, and the percentage of time the trajectory spent in the FT for the seven continental high CN number concentration periods (HCP1-7) are displayed in Fig. S9, and for five continental low CN number concentration periods (LCP1-2, LCP5, and LCP7-8) in Fig. S10. To align with the back-trajectory data points, the CCN number concentrations and aerosol size distributions were averaged
across an hour period. It should be emphasized that the Spearman correlation coefficients vary slightly based on the fit estimates for CN number concentration, and number concentrations in the accumulation and Aitken modes. Thus, the Spearman correlation coefficients shown in Figs. S9 and S10 represent only one of many possible. Nevertheless, as expected, a consistently positive correlation between CCN number concentration and CN number concentration, as well as CCN number concentration and number concentration in the accumulation mode was observed, whereas no significant impact on CCN
number concentration from the number concentrations in Aitken and nucleation mode was revealed. Changes in the fraction spent in the FT, and precipitation showed a positive correlation during some events or negative during others. In general, no consistent relationship between the origin of air masses, aerosol size distribution, fraction of precipitation along the back-trajectory, or time spent in the FT, and the CCN number concentration exists. This could be the case since the particle size distributions and the other metrics did not change statistically significantly during a single event. However, the two metrics
that seem to have the greatest effect on the CCN number concentration in this part of the GBR during continental outflows are CN number concentration and precipitation. The marine periods present no link between the air mass origin and the particle size distribution (Fig. S11 and Fig. S12). This indicates that differences in CN and CCN number concentrations are not related to the air mass origin. Instead, it appears that the variations are caused by the air mass's distance from the continent. This distance determines the relative impact of long-range transported aerosols vs. local aerosol sources. The Spearman correlation
coefficients (Fig. S13 and Fig. S14) again demonstrate a positive correlation between the CCN number concentration and the CN number concentration, as well as the number concentration in the accumulation mode. Additionally, a positive correlation in the Aitken mode could be noticed. Precipitation exhibited a consistent negative influence. Compared to the continental periods, where CCN number concentration affecting factors vary greatly from period to period, marine periods provide a more consistent picture given that marine aerosol populations are more homogeneous in size and composition than continental
aerosol populations. Nonetheless, a larger dataset would be necessary to adequately evaluate correlations between various parameters.

### 3.3 CCN activation ratio

To further understand the impact of CCN number concentrations on cloud formation, CCN activation ratios (CCN/CN) were calculated. CCN/CN is determined by $D_{crit}$ for CCN activation, which is inversely associated with $\kappa$. $D_{crit}$ is shown by vertical

lines in the size distributions in Fig. 5, Fig. S7, Fig. S8, Fig. S11, and Fig. S12 and listed in Table S1. Figure S15 presents the CCN/CN ratios for MP and CP at various SS levels. For all SS levels, the activated aerosol fraction, is larger in the CP than in the MP. However, separating the CCN/CN at the cloud-relevant SS level into the distinct periods demonstrates that the situation is not as straightforward (Fig. 6 and Fig. S16). In general, there is no clear correlation between the air mass pathways and origin and their CCN/CN ratios.

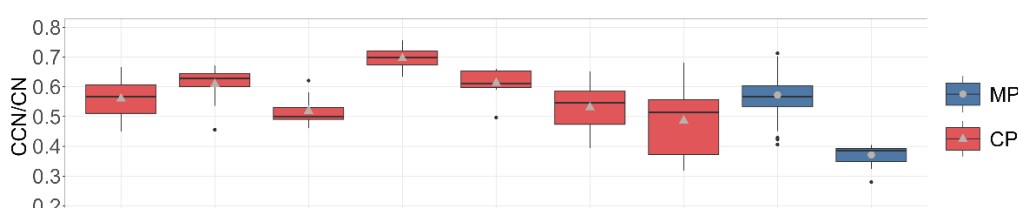

**Figure 6:** CCN activation ratios (CCN/CN) at 0.3 % SS for the MP (blue) and the CP (red) for the high CN number concentration periods (HCP1-7 and HMP1-2). The black horizontal line in a box displays the median of the individual data. The lower and upper hinges represent the 25th and 75th percentiles. The upper and lower whiskers extend from the hinge to the largest or smallest measured values, respectively,
but not more than 1.5 times the difference between the 25th and 75th percentiles. The mean is shown as grey points for the MP and grey triangles for the CP. Outliers are individual data points that fall outside of this range and are color-coded black.

## 4 Conclusion and outlook

In-situ atmospheric data sets were obtained in February and March 2022 to characterise aerosol sources, aerosol size distributions, and CCN properties in the Central and Northern GBR. These measurements have the ability to improve our
understanding on aerosol-cloud interactions in this region.

The CCN concentrations were highly variable during the voyage, with average CCN number concentration for the MP of 194±72 cm$^{-3}$ (median 196 cm$^{-3}$), and 370±93 cm$^{-3}$ (median 358 cm$^{-3}$) for the CP. The higher concentrations in the CP suggest a considerable influence from continental emissions to CCN number concentrations in this region of the GBR.

The hygroscopicity parameter κ suggests an internally mixed composition of nss-sulfates and organics for both the MP and
370 CP, with a higher organic contribution during the CP compared to the MP. Given that continental outflows of both anthropogenic and biogenic origin contain a considerable amount of organic material this is not surprising. In general, the low κ values suggest that organics contribute considerably to CCN in this region of the GBR during both the CP and the MP. The averaged κ values for the MP were 0.188±0.081 (median 0.192), and for the CP 0.293±0.267 (median 0.172). No indicative difference in median κ was observed between the MP and the CP at cloud-relevant SS. Low κ values and greater CCN number
concentrations for sea spray enriched in organics was previously observed by Andreae and Rosenfeld (2008), Ovadnevaite et al. (2011), and Ovadnevaite et al. (2017). This could explain that in our study the CCN number concentration in the CP is higher than in the MP, despite the somewhat lower κ.

The average CN number concentration for the CP was 631±209 cm$^{-3}$ (median 559 cm$^{-3}$), and 387±101 cm$^{-3}$ (median 386 cm$^{-3}$) for the MP. A distinct bimodal aerosol number size distribution, with a clear Aitken and accumulation mode, separated by

the Hoppel minimum, as expected for cloud-processed marine aerosols could be observed during MP. Higher CN number concentrations in the accumulation mode were assumed to be the reason for the higher CCN number concentrations in the CP compared to the MP.

However, the Spearman correlation coefficients for a correlation significance level $p < 0.05$ between the CCN number concentration, CN number concentration inferred from the total fit, number concentration in the accumulation mode, number concentration in the Aitken mode, number concentration in the nucleation mode, averaged precipitation along one trajectory, and the percentage of time the trajectory spent in the FT reveal that no consistent relationship between the origin of air masses, size distribution, fraction of precipitation along the back-trajectory, or time spent in the FT, and the CCN number concentration exists. The two metrics that seem to have the greatest effect on the CCN number concentration in this part of the GBR during continental outflows are CN number concentration and precipitation along the back-trajectory pathway. Future measurements would benefit from co-located measurements of particle number size distributions, CCN number concentrations, and precipitation rates to better understand the impact of local precipitation. Marine periods also indicate a positive correlation with the number concentration in the Aitken and accumulation mode.

The activated aerosol fraction is larger in the CP than in the MP. However, the CCN/CN for the distinct periods are similar in some of the continental and marine periods, and there is no apparent association between air mass paths and origins and their CCN/CN ratios.

The Cooling and Shading subprogram within the RRAP is investigating the feasibility of two independent technologies, MCB and fogging, to minimise the amount of solar radiation reaching the GBR. The data will help to identify sections of the GBR that would benefit the most from MCB. Due to energy and economic constraints, it may be desirable to limit MCB implementation to sub-regions of the GBR. Locations that would be best suited for implementing MCB on the GBR are determined not only by where it is most needed, but also by where it will be most effective. Therefore, to target the optimum increase in cloud albedo from activation of the aerosol into cloud droplets, the background aerosol properties need to be known. The implications for MCB are that if we know the typical number concentration of aerosols in the marine background that can act as CCN and how these vary across the GBR, we can estimate how many cloud seeding particles of a known size need to be injected to increase the cloud droplet number concentration and achieve the required cloud brightening. To achieve a climatological understanding of aerosol and CCN properties over the GBR long-term continuous measurements will be required to capture the variety of atmospheric conditions more comprehensively over the entire latitudinal range of the sparsely observed GBR. This includes both transect voyages as well as stationary observations spanning several bleaching seasons. The results presented here represent a first step over the northern segment of the GBR for which no previous data exist.

**Data availability**

Data for this work can be found on Zenodo (DOI:10.5281/zenodo.15064303).

## Author contribution

E.J.H. analysed the data and prepared the manuscript. J.A. conducted the instrument installation and analysed the data. L.H conducted the instrument installation and remote monitoring throughout the voyage and designed the Cloud-Cube. L.C. conducted the instrument installation. D.P.H. conceived the project, obtained funding, supervised the project and conceived and designed the Cloud-Cube. Z.D.R. supervised the project.

## Conflicts of interest

The authors declare that they have no conflict of interest.

## Acknowledgements

This work is supported by the Reef Restoration and Adaptation Program which is funded by the partnership between the Australian Government's Reef Trust and the Great Barrier Reef Foundation. The authors would like to acknowledge the Traditional Owners of the Great Barrier Reef for permission to undertake research in their traditional sea country. The authors thank the vessel crew who was operating the equipment according to provided instructions and Fabian Marth for his valuable insights and constructive discussions.

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
