# Peer review of "Measurement report: Aerosol and cloud nuclei properties along the Central and Northern Great Barrier Reef – Impact of continental emissions"

_EGUsphere, 2025_

## Referee Comment (RC1)

**Reviewer Comment**

This study investigates aerosol and cloud condensation nuclei (CCN) properties over the Central and Northern Great Barrier Reef during a 2022 research cruise. It finds that continental air masses increase CCN concentrations despite lower aerosol hygroscopicity, largely due to organic emissions. Precipitation history and aerosol source regions notably influence particle properties. These results are important for assessing Marine Cloud Brightening as a climate intervention to protect the reef. Long-term observations are recommended for a full climatological understanding. Measurement data with analysis presented in this paper is meaningful, and the manuscript and figures supporting the points are sufficient for the publication. However, it would be nice to reflect some of points listed below in the manuscript.

1. In the section 2, it seems that a description of the meteorological conditions during the observation period is missing. Since these conditions can also influence aerosol concentration, size distribution and CCN properties, it would be beneficial to include a discussion on this aspect. Even if they are not directly relevant to the main findings, providing this information is essential to help readers better understand the measurement period.

2. The measurement period covered in the study was from February to March. Are there any other measurement results or references from similar locations during different seasons? If available, including discussion or comparison regarding seasonal influences could further enrich the analysis.

3. While the study addresses the precipitation effect on aerosol and CCN properties, it appears that only correlation coefficient analysis is presented. A more detailed case study analysis seems necessary to better understand how precipitation actually impacted on the aerosol characteristics – for example, by comparing aerosol properties before and after precipitation events. Additionally, a more in-depth discussion on how precipitation influences high concentration cases would further strengthen the interpretation of the results.

4. It would be important to discuss whether any new particle formation (NPF) events observed during the measurement period, and whether such events contributed to the observed high aerosol concentrations. In the current manuscript, only an increase in

the nucleation mode is mentioned, but there does not appear to be any explicit discussion of NPF events. Given the size range covered by SEMS, it seems sufficient to detect NPF events, and therefore a more detailed discussion on this point would be valuable.

5. In the supporting information, the method for deriving $\kappa$ value is explained. This approach estimates $\kappa$ value based on size distribution and CCN number concentration, rather than direct measurement such as size-resolved CCN measurements of HTDMA. Since this method inherently involves assumptions about the aerosol mixing state, it would be beneficial to explicitly mention these assumptions and to discuss the associated uncertainties in the estimation method.

6. In Figure 5, the dots representing precipitation events are highly clustered, which makes it difficult to distinguish. It is recommended to consider alternative visualization strategies to improve clarity.

---

## Author Response (AR1)

**Reviewer 1:**

This study investigates aerosol and cloud condensation nuclei (CCN) properties over the Central and Northern Great Barrier Reef during a 2022 research cruise. It finds that continental air masses increase CCN concentrations despite lower aerosol hygroscopicity, largely due to organic emissions. Precipitation history and aerosol source regions notably influence particle properties. These results are important for assessing Marine Cloud Brightening as a climate intervention to protect the reef. Long-term observations are recommended for a full climatological understanding. Measurement data with analysis presented in this paper is meaningful, and the manuscript and figures supporting the points are sufficient for the publication. However, it would be nice to reflect so me of points listed below in the manuscript.

[R1C1] In the section 2, it seems that a description of the meteorological conditions during the observation period is missing. Since these conditions can also influence aerosol concentration, size distribution and CCN properties, it would be beneficial to include a discussion on this aspect. Even if they are not directly relevant to the main findings, providing this information is essential to help readers better understand the measurement period.

*A few sentences "The temperature ranged from 23.28° C to 32.38° C over the course of the measurement period, with an average temperature of 28.09±1.12° C (median 28.28° C). The RH ranged from 43.64 % to 90.64 %, with an average value of 75.54±5.57 % (median 76.64 %). A time series of the temperature and RH for the entire period is provided in Fig. S4." were added to the main manuscript (Line 157-159), the according figure added to the SI, and the section 2.5 renamed into "Air mass categorisation and meteorological conditions".*

[R1C2] The measurement period covered in the study was from February to March. Are there any other measurement results or references from similar locations during different seasons? If available, including discussion or comparison regarding seasonal influences could further enrich the analysis.

*There were other short voyages, such as a transit voyage with the RV Investigator in October 2019. However, the vessel was only passing through the area and spent less than*

*one day there. After removing data influenced by ship exhaust, there were not enough data points for a comprehensive analysis.*

*There was also a voyage on the RV Investigator in October 2016, but the data from that voyage will be presented in a separate publication.*

[R1C3] While the study addresses the precipitation effect on aerosol and CCN properties, it appears that only correlation coefficient analysis is presented. A more detailed case study analysis seems necessary to better understand how precipitation actually impacted on the aerosol characteristics – for example, by comparing aerosol properties before and after precipitation events. Additionally, a more in-depth discussion on how precipitation influences high concentration cases would further strengthen the interpretation of the results.

*During this study, the ship was not stationary for most of the time, and the particle number size distribution was measured at the location of the ship, while precipitation was estimated along the back-trajectory. As a result, a direct comparison between the locally measured size distribution before and after precipitation events is not feasible. We acknowledge that future measurements would benefit from co-located measurements of the particle number size distribution and the precipitation rates to observe the direct impact of precipitation on the particle number size distribution. A sentence on was added to the conclusion and outlook part saying, "Future measurements would benefit from co-located measurements of particle number size distributions, CCN number concentrations, and precipitation rates to better understand the impact of local precipitation." (Line 389-391).*

[R1C4] It would be important to discuss whether any new particle formation (NPF) events observed during the measurement period, and whether such events contributed to the observed high aerosol concentrations. In the current manuscript, only an increase in the nucleation mode is mentioned, but there does not appear to be any explicit discussion of NPF events. Given the size range covered by SEMS, it seems sufficient to detect NPF events, and therefore a more detailed discussion on this point would be valuable.

*The aerosol number size distribution for the entire measurement period was added to the SI as Fig. S6 and a sentence was added to the main manuscript saying, "This can also be*

*observed in the particle size distribution displayed in Fig. S6." (Line 242). An increase of particles in the nucleation mode inarguably increases also the total number concentration, however, as already mentioned in Line 231-232 (now Line 245-246) particles in the nucleation mode are not expected to contribute to CCN. This was also confirmed when comparing the observed nucleation events with the CCN concentration. No correlation between those could be observed (plots not shown). A further detailed discussion on NPF events is thus out of the scope of this measurement report.*

[R1C5] In the supporting information, the method for deriving κ value is explained. This approach estimates κ value based on size distribution and CCN number concentration, rather than direct measurement such as size-resolved CCN measurements of HTDMA. Since this method inherently involves assumptions about the aerosol mixing state, it would be beneficial to explicitly mention these assumptions and to discuss the associated uncertainties in the estimation method.

*A few sentences were added to Line 49-53 in the SI stating "For this calculation, a size-independent homogeneous aerosol composition and internal particle mixing were assumed. This is presumably only valid for background aerosols, while in most cases aerosols of different size have different composition and are not necessarily internally mixed. The uncertainties associated with this method could result in an incorrect estimation of the critical diameter ($D_{crit}$) and thus also of the required SS for CCN activation."*

[R1C6] In Figure 5, the dots representing precipitation events are highly clustered, which makes it difficult to distinguish. It is recommended to consider alternative visualization strategies to improve clarity.

*For better visualisation data points every 12 hours along the back-trajectory are plotted. The original figure was moved to the SI as Fig. S6. A sentence was added to the manuscript (Line 293-295) stating: "For improved visualisation, only data points every 12 hours along the back-trajectories are shown. Hourly resolved back-trajectories are presented in Fig. S6. "*

**Reviewer 2:**

The measurement report by Horchler et al. is reporting aerosol size distributions and their activation properties over the GBR which is valuable and informing. The level of data analysis is sufficient for measurement report and the report can be published after addressing the comments. I am particularly pleased with the authors effort to deconvolve size distributions and to derive Hopple minimum which is very relevant not only for the context of the report, but to all aerosol community.

There are, however, several issues with the reporting content.

[R2C1] First, Hoppel minimum, which is essentially the boundary between the Aitken and Accumulation modes, should be systematically presented along with derived Dcrit by CCN-SMPS method. The authors may not or cannot go explaining what can be learned from the consistency or inconsistency of the two metrics (due to the lack of chemical composition), but at least present a Table of both for notable size distributions or observed events.

*$D_{crit}$ has been added to Figure 3, and the Hoppel minima to Figure 5. The plot captions have been amended accordingly, also in figures in the SI. All Hoppel minima and $D_{crit}$ have been also added to Table S1 in the SI. Additionally, "During the CP, an additional mode at around 30 nm (nucleation mode, dotted orange) was observed. Since the CP represents an average of many different continental outflow events, the Hoppel minimum is not clearly defined in the total lognormal fit (solid black line in Fig. 3). When examining the individual data points, it appears that the CP has a Hoppel minimum at 66.1 nm, which coincides to the MP's minimum at 68.3 nm. The critical diameter ($D_{crit}$) and the Hoppel minimum for MP and CP are listed in Table S1 in the SI." was added (Line 231-236). In Line 292 "... including $D_{crit}$ and, if present, the Hoppel minimum...", in Line 296-297 "The Hoppel minimum and $D_{crit}$ of all periods are additionally listed in Table S1. ", and in Line 350 "...and listed in Table S1." was added. And in-depth discussion of these values and why they are different is beyond the scope of this measurement report.*

[R2C2] Second, there is an inconsistency throughout the text between the average and the median values. Aerosols distribute by log-normal law and the median would be most appropriate, but for the purposes of comparison (or aiding readers) in the measurement

report the authors could present both in all situations, e.g. put either in the brackets next to the other metric.

*Where appropriate, the manuscript was amended to present mean values and in brackets the median values.*

Other comments in their sequence.

Line 115. what was the range of RH during the measurements?

*Please see answer to [R1C1].*

Line 125. In-cloud SS was determined by CCN-SMPS method and/or Hoppel minimum. Why there is a mention of ammonium sulphate? Suppl. Figure S4 does not suggest ammonium sulphate composition either, because kappa 0.2-0.3 for MP/CP is not close to ammonium sulphate kappa of 0.55. Please clarify and explain or amend.

*Thank you for spotting this. This is indeed an error and "for particles composed of ammonium sulfate" was removed from Line 125.*

Line 134. Surprisingly, aethalometer absorption measurements are absent which are crucial in determining and quantifying anthropogenic pollution, especially when dealing with continental outflow. There could have been some interesting insights into kappa relationship with BC if that was measured. Could the authors tell why such a critical instrument was not deployed?

*A Tricolour Absorption Photometer (TAP) from Brechtel was deployed on the voyage. Unfortunately, the instrument stopped working after a couple of days.*

Figure 3. Hoppel minimum exists in CP as well as long as the Aitken and Accumulation modes can be deconvolved. So the Hoppel minimum is fairly similar in both and should be specified.

*Please see answer to [R2C1].*

Line 237. The authors may find a key study by Ovadnevaite et al. 2017 in Nature explaining and supporting their results for possible liquid-liquid phase separation phenomenon. *Thank you for pointing to this valuable study. A sentence addressing liquid-liquid phase separation as an alternative explanation has been added to Line 249-250, saying*

*"Alternatively, liquid-liquid phase separation of organic-rich aerosol particles could have contributed to the enhanced CCN number concentration (Ovadnevaite et al. 2017)."* The corresponding reference was added to the reference list, as well as in Line 192-193 and Line 376.

Figure 5. What can we learn from the discrepancy of critical diameter determined by CCN-SMPS method and Hoppel minimum (intersection of Aitken and Accumulation modes can be considered as Hoppel minimum if not visually present)? Perhaps the discrepancy tells about the heterogeneity of size dependent chemical composition not accounted for by CCN-SMPS method which assumes it being uniform and internally mixed.

*The Hoppel minimum is driven by the supersaturation during cloud processing along the back trajectory, whereas the critical diameter from the CCN measurements is determined by the 0.3% supersaturation set during our measurements. Any difference between the Hoppel minimum and the critical diameter is most likely due to this. A sentence was added saying "Differences between the Hoppel minimum and $D_{crit}$ are most likely caused by the fact that the Hoppel minimum is driven by the SS during cloud processing along the back-trajectory, whereas $D_{crit}$ from the CCN is determined by the set SS during our measurements." (Line 236-238)*